# *OsCSLD1* Mediates NH_4_^+^-Dependent Root Hair Growth Suppression and *AMT1;2* Expression in Rice (*Oryza sativa* L.)

**DOI:** 10.3390/plants11243580

**Published:** 2022-12-19

**Authors:** Sujeevan Rajendran, Chul Min Kim

**Affiliations:** Department of Horticulture Industry, Wonkwang University, Iksan 54538, Republic of Korea

**Keywords:** rice, ammonium, root hair, *CSLD1*, elongation

## Abstract

Root hairs play crucial roles in the roots, including nutrient uptake, water assimilation, and anchorage with soil, along with supporting rhizospheric microorganisms. In rice, ammonia uptake is mediated by a specialized ammonium transporter (*AMT*). *AMT1;1*, *AMT1;2*, and *AMT1;3* have been extensively studied in relation to nitrogen signaling. *Cellulose synthase-like D1* (*CSLD1*) is essential for cell expansion and is highly specific to root hair cells. *csld1* mutants showed successful initiation but failed to elongate. However, when nitrogen was depleted, *csld1* root hairs resumed elongation. Further experiments revealed that in the presence of ammonium (NH_4_^+^), *csld1* roots failed to elongate. *csld1 *elongated normally in the presence of nitrate (NO_3_^−^). Expression analysis showed an increase in root hair-specific *AMT1;2* expression in *csld1*. *CSLD1* was positively co-expressed with *AMT1;2* changing nitrogen concentration in the growth media. *CSLD1* showed increased expression in the presence of both ammonium and nitrate. Methylammonium (MeA) treatment of *CSLD1* overexpression lines suggests that *CSLD1* does not directly participate in nitrogen transport. Further studies on the root hair elongation mutant *sndp1* showed that nitrogen assimilation is unlikely to depend on root hair length. Therefore, these results suggest that *CSLD1* is closely involved in nitrogen-dependent root hair elongation and regulation of *AMT1;2* expression in rice roots.

## 1. Introduction

Root hairs are tube-like outgrowths of root epidermal cells, a functionally essential part of the root system in plants that serve as the interface for nutrient uptake, water assimilation, and anchorage with soil. Root hairs aid these functions by effectively increasing the contact surface of the plant with the soil–particle interface [1]. Root hair cells are characterized by the release of organic compounds (exudates) into the rhizosphere, which can shape the microbial community around the plant root [2]. Root hairs develop from a group of specific root epidermal cells called trichoblasts, in contrast to atrichoblasts, which are not involved in their formation [3]. However, the arrangements of these cells are species-dependent and can be classified into three types. Type I, such as in rice, where trichoblasts and atrichoblasts are arranged randomly to facilitate a plastic response under alternating surrounding conditions. Type II patterning can be seen in grasses such as *Brachypodium*, where trichoblasts are derived from asymmetric cell division, larger cells become non-hair-forming cells and smaller cells live to produce root hair. Type III patterning is widely seen in *Brassicaceae* species such as Arabidopsis, where the hair-forming and non-hair-forming cells are organized as cell files inter-spread with other cell files [1,4,5]. However, cell patterning and cell fate is defined by positional information, not by cell lineage [6]. Root hair development is regulated by genetic and environmental factors. Root hair development occurs in three steps: root hair cell specification, initiation, and elongation. The process of cell growth or elongation is an expansion of the cell wall relative to the protoplast. During expansion, essential components such as cellulose and hemicellulose are synthesized at the plasma membrane. Pectin and xyloglucan were synthesized in the endomembrane. Synthesized materials are transported to the expansion site by vesicles and released (see review) [7]. The first root hairless mutant, *rh2*, which has yet to be cloned, could recover partially when applied with exogenous NAA (α-naphthaleneacetic acid), indicating a shortage in the endogenous auxin [8]. *OsWOX3A* controls root hair formation by regulating the auxin transporters [9]. In contrast, the transcription factor *OsRHL1*, signaling *OsFORMIN HOMOLOGY 1* (*OsFH1*) [10], expansins such as *EXPA17* (*OsEXPA17*) [11], and phosphatidylinositol transfer protein *OsSEC14-nodulin domain-containing protein 1* (*OsSNDP1*) [12] were also found to regulate root hair development.

Plant cell walls mediate essential activities, such as growth, development, and biotic and abiotic responses, and facilitate the cellular communication [13,14,15]. Cell walls are made of complex interactions of biosynthetic enzymes including glycosyl transferase (GT) enzymes that catalyze the formation of glycoside bonds in cell wall glycan polymers [16]. Cellulose, hemicelluloses, and pectins form a complex network in the primary cell walls of most cells to provide mechanical stability [13,17]. *CELLULOSE SYNTHASE- LIKE* (*CSL*) family genes are known to synthesize the backbones of hemicelluloses [18,19,20]. *CSL* genes are a multigene family of proteins that share sequence similarities with cellulose synthase (*CESA*). Several *CSL* genes have been identified in various plant species. However, their role in polysaccharide synthesis is not yet fully understood [21]. Cellulose synthase inhibitors cause the expanding cells to rupture due to the weakness of the expanding cell walls to compensate for the expanding protoplast (see review) [7]. All *CSL* genes in rice are essential for cell wall biosynthesis. *CSLD*s belong to cellulose synthase and cellulose synthase-like gene superfamily (*CESA*/*CSL*). *CSLD1* and *CSLD4* showed specific expression in radicle and plumule, respectively. *CSLD2* expression was observed in all tissues and *CSLD3* showed stamen specific expression [22]. In Arabidopsis, highly coordinated expression of GTs such as *CSLD*s suggests the importance of GTs in cell plant morphogenesis [23]. *CSLD1* mutants exhibited defects in pollen tube [24]. *CSLD1* is also expressed in trichoblasts before root hair growth and is essential for root hair elongation. Ectopic expression and mutant analysis showed that root hair specification was independent of *CSLD1*. Moreover, *OsCSLD1* displayed sequence and mutant phenotypic similarity to *KOJAK*/*AtCSLD3*. *KOJAK* mutants fail to cause root hair elongation by failing to transport cell wall-related polymers to the root hair expansion site [25].

Nitrogen use efficiency is critical for the productivity of rice crop and the environment. Most nitrogen fertilizer recommendations are based on NH_4_^+^ compounds. Developing genotypes with higher nitrogen use efficiency under low nitrogen level is beneficial for the environment and farm economy [26,27]. Root hair formation is highly responsive to nitrogen availability in the surrounding media. Studies related to N-related root hair growth are rapidly emerging [28]. Recently, higher NO_3_^−^ levels were shown to increase root hair density by suppressing trichoblast elongation in Arabidopsis. This is achieved by the involvement of *NRT1;1* and *TGA1/4*, which regulate the root hair-specific gene CPC [29]. Ammonium transporters play a crucial role in nitrogen uptake from the soil solution to the roots in the form of NH_4_^+^. High-affinity ammonium transporters such as *AMT1;1*, *AMT1;2*, and *AMT1;3* are known to mediate important morphological and physiological responses under various levels of ammonium ion concentrations [30,31,32]. Rice contains three *OsAMT2* gene family members: *OsAMT1;1* (identical to *OsAMT1,1*), *OsAMT1;2* (identical to *OsAMT1,3*), and *OsAMT1;3* (identical to *OsAMT1,2*) [31,33,34]. *OsAMT1;1* is highly expressed in shoots and can be induced by soil nitrogen. *OsAMT1;2* and *OsAMT1;3* are strictly expressed in roots. *OsAMT1;2* is induced when ammonium is present and *OsAMT1;3* is repressed by NO_3_^−^, where *OsAMT1;3* acts as a sensor for NH_4_^+^ ions. In contrast, only *AMT1;1* is expressed in the roots in the presence of NO_3_^−^ [31,33]. Ammonium transporters *AMT1;2* were triggered by the introduction of ammonium ions into the roots. In *Arabidopsis*, *CAP1* has shown to regulate root hair tip growth by regulating cytoplasmic Ca^2+^ gradients along root hair cells. Mutants of *CAP1* did not produce normal root hairs on Murashige and Skoog (MS) medium. However, they resumed regular root hair growth when NH_4_^+^ was depleted [35]. Moreover, *AMT1;2* expression was increased under phosphorus deficiency which indicates a relationship between nutrient availability and *AMT1*s [36]. Root-hair-specific *oscsld1* mutants also exhibited regular root hair initiation and failed to elongate root hair under regular conditions. In this study, we established a relationship between NH_4_^+^ and NO_3_^−^-mediated *CSLD1*-dependent root hair elongation in rice (*Oryza sativa* L.).

## 2. Results

### 2.1. NH_4_^+^ Suppresses Root Hair Elongation of csld1

Since *csld1* mutants successfully initiated root hairs in the seminal roots and only failed to elongate root hairs [25], *csld1* mutants were grown without NH_4_^+^ and/or NO_3_^−^ to examine any effect of *CSLD1* on nitrogen-mediated root hair elongation. Initially, root hairs failed to elongate in half-MS media and without CaCl_2_ (Figure 1A and B) but *csld1* root hairs demonstrated an increase in length when the supply of NO_3_^−^ was depleted (Figure 1C). When NH_4_^+^ was absent, the roots of *csld1* recovered their length to the wild-type root hairs (Figure 1D).

To verify the effect of NH_4_^+^ on *csld1* root hairs, mutants were grown in NH_4_^+^-deficient media. In the absence of NH_4_^+^ alone, root hair length of *csld1* significantly increased relative to the standard half-MS media counterpart. However, the wild-type roots did not show significant differences between standard and modified half-MS media (Figure 2A,B). Investigation of the expression levels of *CSLD* genes by qRT-PCR revealed that in wild-type roots, expression of *CSLD3* and *CSLD4* expression was increased in the presence of NH_4_^+^. In *csld1* mutants, the expression of *CSLD2*, *CSLD3,* and *CSLD4* were significantly increased under normal conditions. In the presence of NH_4_^+^, the *csld1* mutants did not show any significant differences in expression in any other *CSLD* genes (Figure 2C).

To distinguish the effects of different nitrogen sources on root hair elongation, *csld1* mutants were grown in KNO_3_ and NH_4_NO_3_. In the presence of KNO_3_, *csld1* produced root hairs that were similar in lengths that of the wild-type root hairs. On the other hand, when NH_4_NO_3_ was added to the media, *csld1* mutants showed significant reduction in root hair length (Figure 3A,B). It is noteworthy that the reduction of root hair length in *csld1* mutants in the presence of NH_4_NO_3_ was similar to the root hair length of the wild-type root hair grown in KCl.

Following the observation from KNO_3_ and NH_4_NO_3_ treatments, wild-type and mutant seedlings were grown in NH_4_Cl or KNO_3_ to distinguish the effect of both NH_4_^+^ and NO_3_^−^. In the presence of NH_4_^+^
*csld1* mutants showed significant reduction in root hair length when compared with the wild-type root hairs. However, in the presence of NO_3_^−^, both wild-type and *csld1* root hairs elongated up to twice that of the wild-type root hairs supplemented with NH_4_^+^ and showed significant difference (Figure 4A,B), suggesting that *csld1* mutants are not sensitive to NO_3_^−^ but NH_4_^+^ and NO_3_^−^ have more influence on root hair elongation than NH_4_^+^.

### 2.2. CSLD1 and AMT1:2 Show Close Relationship in Activity under Different NH_4_^+^ Concentrations

Differential expression of root specific ammonium transporters such as *AMT1;2* under different nutrient states were observed in previous studies. To identify any relationship between *CSLD1* and the three *AMT1*s, the expression levels of *AMT1* genes were examined in WT and *csld1* mutants. In *csld1* mutants, only *AMT1;2* showed significant increase in expression compared with the wild-type, indicating a negative relationship between *AMT1;2* and *CSLD1* (Figure 5A).

Under nitrogen starvation, Arabidopsis root hairs show an increased length [37], suggesting a possible significant correlation between nitrogen signaling and *csld1* expression. Expression analyses for *CSLD1* and *AMT1;2* were performed to determine their expression levels under different nitrogen levels. When seedlings were transplanted to 1 mM NH_4_^+^ during nitrogen starvation, the initial expression levels of *CSLD1* and *AMT1;2* were similar. However, *AMT1;2* expression was significantly higher than *CSLD1* 3 h after transfer. After 6 h, both *CSLD1* and *AMT1;2* expression levels decreased, while *AMT1;2* expression was maintained at higher levels and *CSLD1* expression was significantly reduced (Figure 5B). However, *CSLD1* and *AMT1;2* expression levels increased simultaneously, with significantly (but only diverging in one point i.e., 48 h) increasing expression levels when seedlings were transferred from high to low NH_4_^+^ levels (Figure 5C). To visualize the expression of *CSLD1* in trichoblasts in the presence of NH_4_^+^, GUS expression patterns were observed under *CSLD1::Ds/CSLD1::Ds* (*csld1*) background. When NH_4_Cl or KNO_3_ was applied to 1/10 Johnson’s solution, GUS expression was increased in trichoblasts relative to the controls (KCl), indicating the induction of *CSLD1* expression in the presence of NH_4_^+^ and NO_3_^−^ (Figure 6).

### 2.3. Ammonium Uptake Is not Directly Correlated with Root Hair Length

Similar to *oscsld1* root hairs, *ossndp1* mutants also exhibited reduced root hair length caused by the depolarization of root hair growth orientation (Figure 7A) [12]. In order to examine the impact of root hair length and roles of *OsCSLD1* and *OsSNDP1* mutants related to nitrogen transport, each mutant was treated with the toxic analog of ammonia, methyl ammonium (MeA) in MS media. In the absence of MeA, both *csld1* and *sndp1* seminal root lengths were similar but significantly shorter than the wild-type. However, in the presence of MeA, *csld1* seminal roots resisted the effects of MeA. On the other hand, WT and *sndp1* seminal roots were susceptible to MeA and showed reduction in root length (Figure 7B,C). When examined for ammonium concentration in roots in MS media, shorter *sndp1* roots showed similar concentrations to that of the wild-type. The *csld1* roots contained less ammonium than the WT and *sndp1* (Figure 7D).

### 2.4. CSLD1 Is not Directly Involved in Ammonium Uptake but Involved in Ammonium Response

*CSL*s are characterized to directly involve in cell wall biosynthesis and patterning. On the other hand, *AMT*s are the primary transporters of ammonia and nitrate. In order to verify the effect of NH_4_^+^ or NO_3_^−^ on root hair growth in *AMT1 RNAi* mutants in which *AMT1;1*, *AMT1;2*, and *AMT1;3* were suppressed, seedlings were germinated and grown in half-MS medium with 0.1 mM NH_4_^+^ or NO_3_^−^ (Figure 8A). Mutants showed significantly strong reductions in root hair length and density (Figure 8B,C). In contrast, seedlings grown in 0.1 mM NO_3_^−^ failed to show any significant difference in root hair length and density (Figure 8D–E).

To further describe the role of *CSLD1* in ammonium signaling, *CSLD1:OX* (Figure 9A) seedlings were grown in methylammonia (MeA). *CSLD1:OX* lines were sensitive to MeA showing reduction in length and *csld1* seminal roots showed no response to MeA when compared with wild-type seminal roots (Figure 9B,D). However, *AMT1;2:OX* lines showed a dramatic reduction in seminal root length relative to the wild-type in the presence of MeA (Figure 9C,E). This observation was further validated when *csld1* and *CSLD1:OX* seedlings were grown in MeA where *csld1* seminal roots showed relatively long lengths under increasing MeA concentration (Appendix A). This indicates a strong resistance of *csld1* for MeA and enhanced susceptibility of *CSLD1:OX*.

## 3. Discussion

Among all the mineral necessities plants have developed to adopt, nitrogen plays a key role in all kinds of plant growth and development [38,39,40,41,42]. Among the major N sources in soil, such as nitrate and ammonia, the former tends to be most abundant in aerobic environments, and the latter is abundant in flooded conditions [43]. Higher levels of ammonia in anaerobic soils are highly toxic to plants. Rice plants have developed a well-regulated equilibrium of direct ammonia uptake mechanisms using ammonium transporters (*AMT*) to avoid the toxic effects of ammonia accumulation [44].

Initial experiments with *csld1*, NH_4_^+^, and NO_3_^−^ suggested that ammonium ions significantly suppressed root hair elongation and nitrate ions enhanced the elongation of root hair even in the presence of ammonium ions, indicating that *CSLD1* promotes root elongation in the presence of NH_4_^+^ rather than NO_3_^−^ (Figure 1 and Figure 2A,B). A similar observation was observed in Arabidopsis *Atrop11^CA^* mutants [45]. *CSLD*s are essential for root hair and pollen tube growth in Arabidopsis and synthesize polysaccharides essential for the cell wall structure in tip-growing cells [24]. Expression analysis of other *CSLD* genes suggested that *CSLD1*, *CSLD3*, and *CSLD4* showed increased activity in the absence of NH_4_^+^. In *csld1* mutants, *CSLD2*, *CSLD3*, and *CSLD4* showed increased expression but reduced expression when NH_4_^+^ was absent (Figure 2C). Based on these results, it is highly likely that *CSLD1* promotes the expression of other *CSLDs* when NH_4_^+^ was absent and suppresses other *CSLDs* when ammonia is present. Moreover, it is possible that *CSLD1* does not conduct a regulative process alone. Root morphology can be shaped by the direct and indirect effects of NH_4_^+^ and NO_3_^−^ [46]. When NH_4_^+^ and NO_3_^−^ were present in the media, *csld1* mutants showed significant reduction in root hair length. When NO_3_^−^ was present, root hair length was similar to the wild-type. When NH_4_^+^ showed strong suppression of root hair length on *csld1* mutants, wild-type root hair also showed significant reduction in length but not up to the degree of *csld1* root hair length (Figure 3 and Figure 4A,B). It is evident that both NH_4_^+^ and NO_3_^−^ produce opposite signals for root hair elongation where NH_4_^+^ results in suppressing signals and NO_3_^−^ results in promoting signals. Both signals converge into the same regulatory pathway to elongate root hair in a dose-dependent manner.

*AMT1*s act as sensors for ammonium sources available in the media [47]. *AMT1;1*, *AMT1;2*, and *AMT1;3* are the most essential physiological and morphological responses to external N signals. Overexpression of *AMT1;1* presented improved growth and yield [33]. In contrast, overexpression of *AMT1;3* reduced the performance [48]. Current experiments with *AMT1* mutants revealed that the mutants had poorly elongated roots in the presence of NH_4_^+^. However, these mutants did not respond to NO_3_^−^. *Csld1* mutants showed increased expression of *AMT1;2*, suggesting that *CSLD1* is a negative regulator of root hair-specific *AMT1;2* expression. The expression dynamics of *CSLD1* and *AMT1;2* showed similar patterns. When plants were exposed to NH_4_^+^ after N starvation, both *CSLD1* and *AMT1;2* showed higher activity indicating *CSLD1* and *AMT1;2* coupled response in progress. Similar patterns were observed for *CSLD1* and *AMT1;2* when plants were transferred from high-to low-N medium, where the expression progressively increased up to 72 h after transfer.

Experiments on roots with GUS staining revealed that when presented with NH_4_^+^ or NO_3_^−^, *CSLD1* showed activity limited to trichoblast cells. This observation is in agreement with previous studies in regular conditions, *CSLD1* is expressed in root hair cells [25]. Unlike *CSLD1*, the short root hair mutant *sndp1* [12] showed hindered root hair elongation when grown in MeA media and *csld1* showed strong resistance to MeA treatment relative to wild-type. This suggests that *CSLD1* is involved in the ammonia response process. It is noteworthy that *sndp1* had ammonium concentrations similar to those of the wild-type. *csld1* had lower ammonium concentrations, implying that root hair length does not affect ammonium content or ammonium absorption capacity of root hair.

The toxic ammonia analog, methyl-ammonium (MeA), can be used to identify ammonium transport [49,50]. Overexpression of ammonium-inducible *AMT1;2* is highly sensitive to MeA. *CSLD1* overexpressing lines showed higher degree of susceptibility to MeA and *csld1* mutants showed significant resistance to MeA, indicating that *CSLD1* is not a part of ammonium uptake (Figure 9). These observations were further validated when *csld1* and *CSLD1:OX* lines were grown in the presence of MeA. In this case, *csld1* seminal roots remained longer than wild-type and *CSLD1:OX* roots when exposed to increasing concentrations of MeA (Appendix A). The results from these experiments suggest that *CSLD1* is involved in the response to NH_4_^+^-triggered root hair elongation. These results also indicate that, unlike *AMT1*s, *CSLD1* is not directly involved in N signaling in rice. Further studies are necessary to explore the possibility of *CSLD1* interacting with *AMT1*s and the role of N signals in the dynamics of *AMT1s* and *CSLD1* related to root hair elongation. Current experiments suggest that the length of root hair is not related to nitrogen uptake in in vitro conditions. However, the conditions in soil can be different and root hair length may play a significant role in nitrogen uptake in soil. Studies must be also done to explore possible tools increase the nitrogen use efficiency by varying root hair length.

## 4. Materials and methods

### 4.1. Mutant Materials

A large-scale Ds transposon population was generated by regenerating gene trap Ds starter lines transferred by Ac via tissue culture [51]. Previously characterized csdl1 and sndp1 seeds from the gene trap mutants were kindly provided by Han’s lab, Gyeongsang University, Republic of Korea [12,25].

### 4.2. Plant Materials and Growth Conditions

The japonica cultivar Dongjin (WT; Wild-type), csld1, 35s::CSLD1, three 5′ *AMT1* RNAi lines (3-1 and 5-2), and sndp1 were used in the experiments. Rice seeds were surface-sterilized with 0.05% SPORTEX and germinated for three days in the dark. Uniformly germinated seedlings were selected and cultured hydroponically in different nutrient solutions [¼ MS (Murashige and Skoog), ¼ KB (Kimura B), and ¼ NS (Nutrient Solution)] containing NH_4_^+^ or NO_3_^−^ as the sole nitrogen source. Media compositions were obtained from previous studies [46,52,53]. Hydroponic nutrient solutions were replaced with fresh media every two days for two weeks. Roots were examined after culturing for 14 days in a growth chamber under the following conditions: 16/8 h light/dark, light intensity 280 μmol m^−2^ s^−1^, temperature 26 °C/18 °C, and 70% humidity. To measure the expression levels of three *OsAMT1* (*1;1*, *1;2*, and *1;3*) and four CSLD (CSLD1, CSLD2, CSLD3, and CSLD4) genes, *WT and csld1*, seedlings were grown hydroponically in ¼ nutrient medium supplemented with 0.5 mM NO_3_^−^ or 0.5 mM NH_4_^+^ for seven days. Total cellular RNAs were extracted from roots. For the media shift assays, germinated seeds were cultured in modified ¼ NS containing 0.1 mM NH_4_NO_3_ for seven days. The samples were transferred and cultured in a nutrient solution containing no NH_4_NO_3_ for nitrogen starvation and for moving low to high nitrogen medium. Germinated seeds were first grown in 0.1 mM NH_4_NO_3_, moved to 1 mM, and 10 mM NH_4_NO_3_ containing media after seven days. The same solutions were replaced with fresh medium every two days.

### 4.3. Isolation of Full-Length cDNA of CSLD1 and Agrobacterium Transformation

The 4.0-kb full-length *OsCSLD1* cDNA was isolated by PCR amplification using primers *OsCSLD1*-5#SpeI (ACTAGTATGGCGTCGAAGGGCATCCTC-AAG) and *OsCSLD1-3*#SpeI (ACTAGTCCAGGGGAAAGAGAAGGATCC-TCC). The PCR product was ligated into a pGEM-T vector (Promega; Madison, WI, USA) and sequenced. A 3.4-kb fragment was excised from the vector by SpeI digestion and ligated into the corresponding site of pCAMBIA1302. Full-length OsCSLD1 cDNA was fused with GFP at its 3′ end and expressed under the cauliflower mosaic virus 35S promoter and nopaline synthase 3′ terminator. Calli of rice (Dongjin) were transformed with Agrobacterium LEA4404 carrying pCAMBIA (35S::OsCSLD1). Rice calli (Dongjin) were transformed with T-DNA carrying the hygromycin phosphotransferase gene as previously described method [54], with slight modifications.

### 4.4. Construction of AMT1 RNAi Vector

To generate *AMT1;1* RNAi transgenic plants, 5′ and 3′ fragments of the *AMT1;1* ORF were amplified using the following primer sets: Ri5-F (gagctcggtaccctcgccgcgcacgtcatccag) and Ri5-R (gaattcctgcaggcatgtgcttgaggccgaaga) and Ri3-F (gagctcggtaccctcgcggcgcacatcgtgcag) and Ri3-R (gaattcctgcagttacacttggttgttgctgtt), respectively. The PCR products were digested and cloned into *Eco*RI and *Sac*I sites for sense orientation insertion and *Kpn*I and *Xho*I sites for antisense orientation insertion in a pBluscript-catalase intron vector. After sequencing, whole inserts were cloned into *Sac*I and *Kpn*I sites of PGA1611 binary vector.

### 4.5. Generation of AMT1 RNAi Transgenic Lines

*AMT1 RNAi* transgenic lines were generated from japonica rice cultivar “Dongjin” via Agrobacterium (LBA4404 strain)-mediated transformation using calli derived from dry seeds [55]. The following transgenic lines were selected and propagated: 5′ AMT1 RNAi lines 5-2 (Ri 5-2 and Ri 3-1 respectively).

### 4.6. Generation of Transgenic Overexpression Plants

To generate plants overexpressing AMT1;2, ORFs were cloned into BamHI and HindIII restriction sites of the pGA1611 binary vector to produce pGA1611-AMT1;2, in which AMT1;2 was expressed under the control of the UBIQUITIN promoter. Rice calli were transformed with pGA1611-AMT1;2 using Agrobacterium-mediated transformation.

### 4.7. Histochemical Analysis

β-Glucuronidase (GUS) activity was visualized by incubating 5-bromo-4-chloro-3-indoyl glucuronide solution (1 mg mL^− 1^). The solution contained 50 mM sodium phosphate buffer (pH 7.0), 10 mM EDTA, 0.1% Triton, 2 mM potassium ferrocyanide, and 200 mg/mL chloramphenicol. Samples were incubated at 37 °C for two days in the dark and dehydrated in a 30 to 70% graded ethanol series. GUS-stained samples were dehydrated and embedded in paraffin wax using the method described above. For toluidine blue staining, roots grown on 0.5 X MS medium were stained with 0.1% aqueous toluidine blue for 2 min and washed with distilled water. The roots on the slides were inspected under a light microscope [25].

### 4.8. RNA Extraction and qRT-PCR

Total cellular RNA was isolated using a Qiagen RNeasy Mini Kit (Qiagen, Valencia, CA, USA) following the manufacturer’s instructions. RNA concentration was measured using a NanoDrop ND-1000 spectrophotometer. The samples were treated with RQ-RNase-free DNase (Promega, Madison, WI, USA). An RNaseH (Toyobo, https://www.toyobo-global.com/) reverse transcription kit was used to synthesize cDNA, according to the manufacturer’s instructions (Promega). qRT-PCR was performed using iQ SYBR Green Supermix (Bio-Rad, Hercules, CA, USA). Amplification and quantification were performed with gene-specific primers using CFX Manager software (Bio-Rad), and values were normalized against internal *UBIQ1*. Three biological and two technical replicates were used for each analysis [46]. All the primers used for qRT-PCR are listed in Appendix A.

### 4.9. RT-PCR Analysis

To detect OsCSLD1 cDNA from the samples, gene-specific primers (forward 5-TCGCCGCCGAACAAGATC-3 and reverse 5-CGGACCACTTGATCTCCAG-3) were used. RT-PCR was performed at 94 °C for 5 min, followed by 25 cycles at 94 °C for 30 s, 58 °C for 30 s, and 72 °C for 1 min. The PCR products were fractionated on agarose gels, stained with ethidium bromide, blotted to Hybond N1 (Amersham Pharmacia Biotech, Amersham, United Kingdom), and hybridized with a 32P-labeled *OsCSLD1* probe [25].

### 4.10. Root Hair Length Measurements

Five-day-old roots of rice plants grown in 1/2 MS solution were observed. The root images were captured with an Olympus SZX12 stereomicroscope system. Images were processed and analyzed using the ImageJ software version 1.53k (National institute of health, Bethesda, MD, USA, http://rsbweb.nih.gov/ij/).

### 4.11. Ammonium Content Determination

Determination of NH_4_^+^ content in the roots by enzymatic digestion was performed using an F-kit (Roche) following the manufacturer’s instructions [56].

### 4.12. Methylammonium (MeA) Treatment

Evenly germinated seeds were hydroponically grown in modified full nutrient (FN) medium (2 mM NH_4_NO_3_, 1 mM KH_2_PO_4_, 1 mM MgSO_4_, 250 mM K_2_SO_4_, 250 mM CaCl_2_, 100 mM NaFe-EDTA, 50 mM KCl, 50 mM H_3_BO_3_, 5 mM MnSO_4_, 1 mM ZnSO_4_, 1 mM CuSO_4_, 1 mM NaMoO_4_, and 1 mM MES, pH 5.8 [KOH]) [57] supplemented with different concentrations of MeA (0, 1, 2.5, and 5.0 mM). Root length, shoot height, and dry weight were analyzed after sampling for 10 days.

### 4.13. Statistical Analysis

All experiments were completely randomized and at least eight replicates were used for the quantitative experiments. Data shown in the figures are expressed in average ±SD (standard deviations) of each group. Quantitative data were analyzed using the JMP 16.0.0 software package (SAS Institute, Cary, NC, USA). All statistical comparisons were performed using the “Fit Y by X” function. A one-way ANOVA followed by Tukey HSD was used to visualize the significance among the experimental groups means at the probability level of *p* < 0.05.

## 5. Conclusions

*CSLD1* mediates ammonia-triggered root hair elongation in rice. Moreover, *CSLD1* also suppresses root hair-specific *AMT1;2* and coexpressed with *AMT1;2* when nitrogen concentration regimes change. The expression of *CSLD1* is confined to trichoblast cells and occurs under both ammonia and nitrate signals. Results with short root hair mutants, such as *sndp1*, showed that root hair length did not affect the absorption of nitrogen in root hairs. Further experiments using MeA suggested that *CSLD1* does not directly participate in ammonium absorption. However, it is noteworthy that the mutants and overexpression lines of *CSLD1* showed minor differences in seminal root length. These results suggest that NH_4_^+^ can elongate root hair via *CSLD1*. Further studies are required to understand the mechanism of ammonia-triggered *CSLD1*-mediated root hair elongation.

## Figures and Tables

**Figure 1 plants-11-03580-f001:**
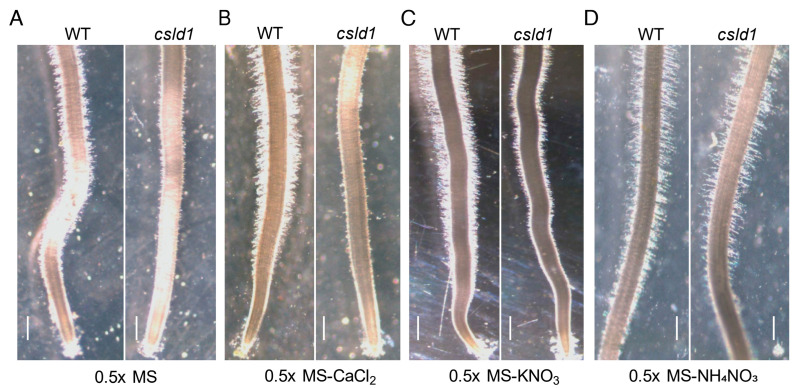
The *csld1* mutants were recovered under nitrogen depletion. Root hair length of wild-type (WT) and *oscsld1 RNAi* mutants grown in (**A**) Half-MS medium, (**B**) Half-MS medium without CaCl_2_, (**C**) Half-MS medium without KNO_3_, and (**D**) Half-MS medium without NH_4_NO_3_. Scale bars = 500 μm.

**Figure 2 plants-11-03580-f002:**
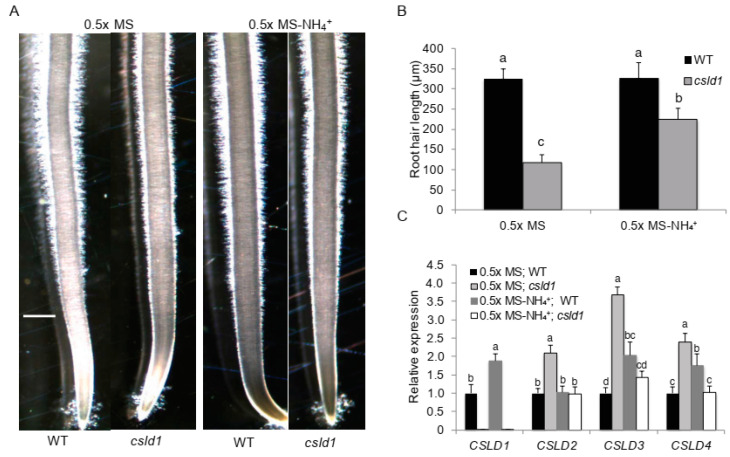
*CSLD1* mutants showed reduction in root hair length and altered the expressions of *CSLD*2/3 in the absence of NH_4_^+^. (**A**) and (**B**) Root hair length of WT and *oscsld1* mutants in MS and modified MS medium without NH_4_^+^. (**C**) Relative expression of *CSLD1*, *CSLD2*, *CSLD3*, and *CSLD4* in WT and *oscsld1* mutants in MS and modified MS medium without NH_4_^+^. Scale bars = 500 μm. Bar graphs show mean values ± standard error of means (*n* = 9). Different letters indicate significant differences between groups according to Tukey’s HSD test (*p* < 0.05).

**Figure 3 plants-11-03580-f003:**
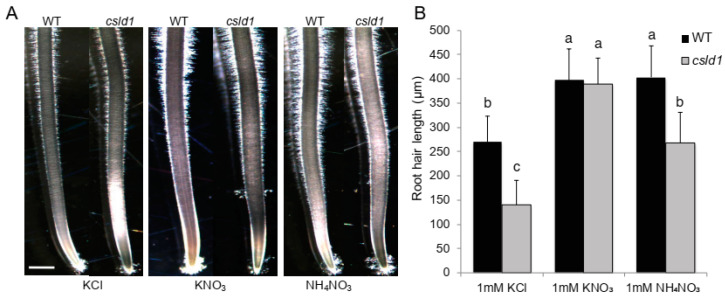
*csld1* root hair failed to elongate in the presence of NH_4_^+^. (**A**) Seminal roots of wild-type and *csld1* mutants grown in 1 mM KCl, KNO_3_, and NH_4_NO_3_, respectively, and (**B**) quantification of root hair length. Scale bars = 500 μm. Bar graph show mean values ± standard error of means (*n* = 8). Different letters indicate significant differences between groups according to Tukey’s HSD test (*p* < 0.05).

**Figure 4 plants-11-03580-f004:**
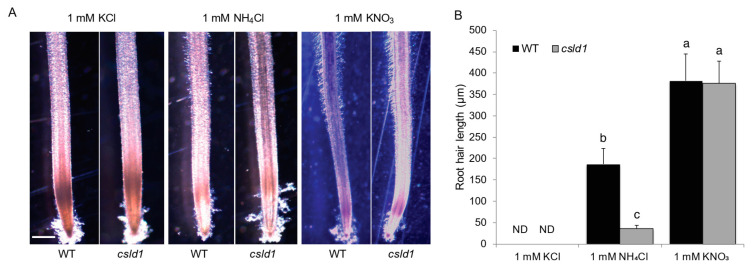
*CSLD1* promotes root hair elongation in the presence of NH_4_^+^. (**A**) Seminal roots of wild-type and *csld1* mutants grown in 1 mM KCl, KNO_3_, and NH_4_NO_3_ and (**B**) quantification of root hair length. Scale bars = 500 μm. Bar graph show mean values ± standard error of means (*n* = 8). Different letters indicate significant differences between groups according to Tukey’s HSD test (*p* < 0.05).

**Figure 5 plants-11-03580-f005:**
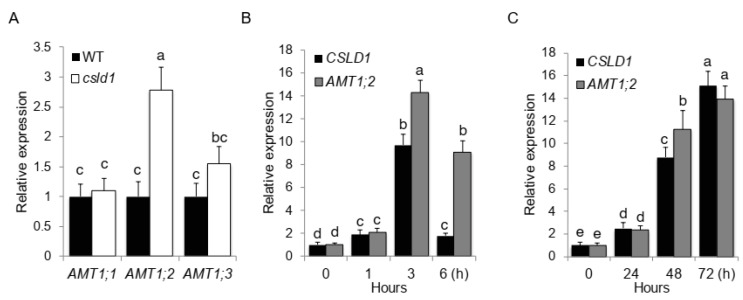
Ammonium-dependent expression patterns of *CSLD1* and *AMT1;2*. (**A**) Relative expression patterns of *AMT1;1*, *AMT1;2*, and *AMT1;3* in root hair in *csld1* mutants. (**B**) Expression pattern of *CSLD1* and *AMT1;2* when transferred from nitrogen starvation to 1 mM NH_4_NO_3_ and (**C**) when transferred from high (10 mM) to low (1 mM) concentration of NH_4_NO_3_. Bar graphs show mean values ± standard error of means (n = 3). Different letters indicate significant differences between groups according to Tukey’s HSD test (*p* < 0.05).

**Figure 6 plants-11-03580-f006:**
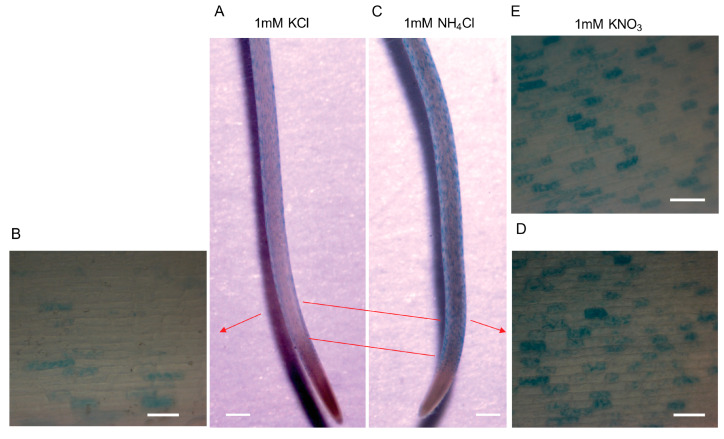
GUS patterns in *CSLD1::Ds/CSLD1::Ds* (*csld1*) roots in different nutrient supplementations. (**A**) The GUS staining of the whole seminal root supplemented with 1 mM KCl and (**B**) 100× magnification of KCl-treated seminal root trichoblast cells. (**C**) Staining of seminal root supplemented with 1 mM NH_4_Cl and (**D**) 100× magnification of NH_4_Cl-treated seminal root trichoblast cells. (**E**) 100× magnification of KNO_3_-treated seminal root trichoblast cells. Red bars indicate the root sections which were used to capture magnified images. Scale bars: **A**,**C**; 500 μm and **B**,**D**,**E**; 50 μm.

**Figure 7 plants-11-03580-f007:**
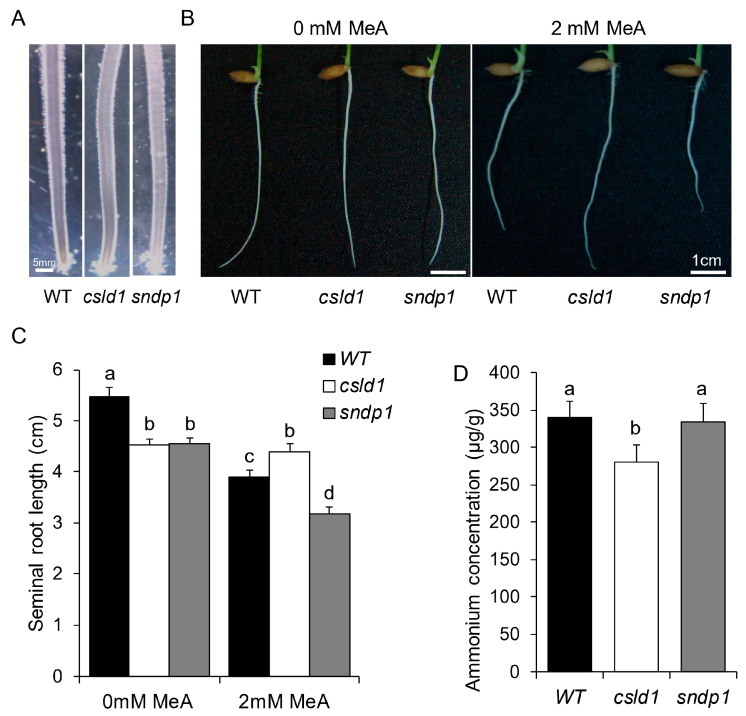
Differences in root hair length did not affect the nitrogen concentration in roots. (**A**) Seminal roots of wild-type, *csld1*, and *sndp1* three days after germination. (**B**) Three-day-old seedlings grown in MS-NH_4_NO_3_ with or without Methyl-ammonium (MeA). (**C**) seminal root lengths of wild-type, *csld1* and *sndp1* seminal roots with or without methyl-ammonium (MeA) and (**D**) ammonium concentration of wild-type, *csld1* and *sndp1* seminal roots. Bar graphs show mean values ± standard error of means (*n* = 8–12). Different letters indicate significant differences between groups according to Tukey’s HSD test (*p* < 0.05).

**Figure 8 plants-11-03580-f008:**
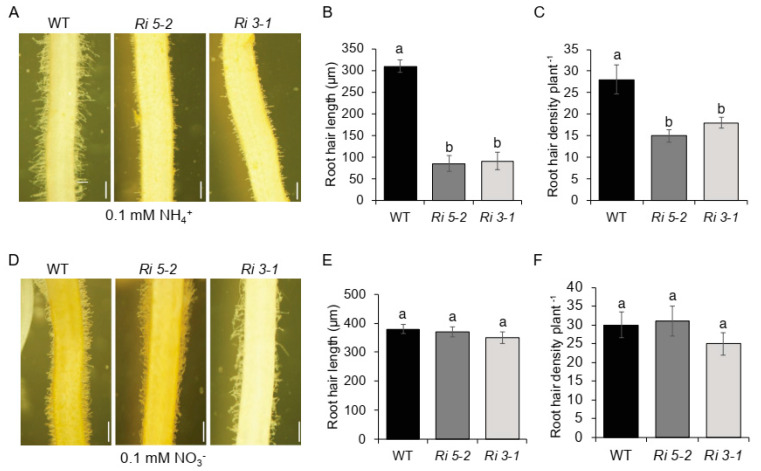
*AMT1 RNAi* displays sensitivity to NH_4_^+^ in MS media but not to NO_3_^−^. *AMT1 RNAi* mutants are cultured with 0.1 mM NH_4_^+^. (**A**) Root hair length and (**B**) root hair density of wild-type (WT) *RNAi 5-2* (*Ri 5–2*) and *RNAi 3-1* (*Ri 3–1*) (**C**) when cultured with 0.1 mM NH_4_^+^. *AMT1 RNAi* mutants cultured with 0.1 mM NO_3_^−^. (**D**) Root hair length (**E**) and root hair density of wild-type (WT) *RNAi 5-2* (*Ri 5–2*) and *RNAi 3-1* (*Ri 3–1*) (f) when cultured with 0.1 mM NO_3_^−^. Scale bars = 500 μm. Bar graphs show mean values ± standard error of means (*n* = 10). Different letters indicate significant differences between groups according to Tukey’s HSD test (*p* < 0.05).

**Figure 9 plants-11-03580-f009:**
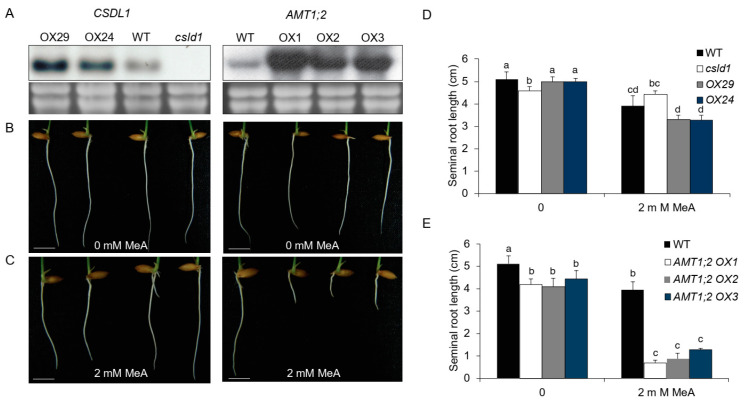
*AMT1;2* overexpressing lines are susceptible to methyl-ammonium (MeA). (**A**) Expression of *CSLD1* overexpressing and mutant lines (left) and *AMT1;2* overexpressing lines (right). (**B**) *CSLD1:OX* lines and *AMT1;2:OX* seeds grown in MS-NH_4_ media without MeA, (**C**) with MeA, (**D**) seminal root length of *csld1* and *CSLD1:OX* seedlings grown in MS-NH_4_ media with/without MeA, and (**E**) seminal root length of *AMT1;2:OX* seedlings grown in MS- NH_4_ media with/without MeA. Scale bars = 1 cm. Bar graphs show mean values ± standard error of means (*n* = 8). Different letters indicate significant differences between groups according to Tukey’s HSD test (*p* < 0.05).

## Data Availability

Data are available by contacting the corresponding author.

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
