# Peer review of "OsCSLD1 Mediates NH4+-Dependent Root Hair Growth Suppression and AMT1;2 Expression in Rice (Oryza sativa L.)"

_plants, 2022, doi:10.3390/plants11243580_

Round 1
Reviewer 1 Report
Many thanks to the authors for their efforts in this manuscript, but I have a few notes:
The introduction should be refined and AMT and CSLD should be included
Materials and Methods
Sections 4.7., 4.8. and 4. 9 ; add references
Line 40: correct “Brasicaceae” to “Brassicaceae”
Line 59: (Yoo, Cho, and Paek, 2013), Give a reference number
Line 284: write (Oryza sativa) in italic
Line 295: (Fig. S1a) is missed
In Figure 1, add the tags a,b, c and d to the images
Enter a title for the table in the supplementary data
In the reference list, the number of each duplicate reference. Remove the duplicate number
Author Response
From,
Prof. Chul Min Kim
Plant Molecular Breeding Laboratory,
Department of Horticulture, Wonkwang University,
54538 Iksan,
Republic of Korea.
Through,
Ms. Nora Huang,
Plants editorial,
MDPI AG, St. Alban-Anlage 66
4052 Basel,
Switzerland.
To,
The editor,
Plants editorial,
MDPI AG, St. Alban-Anlage 66
4052 Basel,
Switzerland.
Dear editor,
I would like to extend my sincere thanks to the reviewers for critically examining the manuscript, which gave us the opportunity to improve the quality of our work by identifying critical points. We were able to make changes and corrections to the manuscript and respond to the comments from the respected editors. Please check the point-by-point response given below.
Review 1
Many thanks to the authors for their efforts in this manuscript, but I have a few notes:
Comment: The introduction should be refined and AMT and CSLD should be included
Response –Agreed. Introduction was revised and more information on AMTs, CLSDs and root hair has been given
Materials and methods
Sections 4.7., 4.8. and 4. 9 ; add references
Response –Agreed. Thank you for pointing this out. References were added according to reviewer comments
Line 40: correct “Brasicaceae” to “Brassicaceae”
Response – Agreed. Thank you for pointing this out. Corrected to Brassicaceae
Line 59: (Yoo, Cho, and Paek, 2013), Give a reference number
Response - Agreed. Thank you for pointing this out. Reference was corrected
Line 284: write (Oryza sativa) in italic
Response - Agreed. Thank you for pointing this out. Text formatting double checked
Line 295: (Fig. S1a) is missed
Response - Agreed. Thank you for pointing this out. This is a misplacement of text Removed.
In Figure 1, add the tags a,b, c and d to the images
Response - Agreed. Thank you for pointing this out. Figure Labels added
Enter a title for the table in the supplementary data
Response - Agreed. Thank you for pointing this out. Table title revised
In the reference list, the number of each duplicate reference. Remove the duplicate number
Response –Thank you for the comment. Duplicates checked by software and manually.
We have made relevant corrections with track changes and attached the corrected manuscript. We are eagerly waiting for the response from the respectable reviewers soon.
Sincerely,
Chul Min Kim

Reviewer 2 Report
In this manuscript (PLANTS-1989005: OsCSLD1 Mediates NH4+-Dependent Root Hair Growth Suppression and AMT1;2 Expression in Rice (Oryza sativa L.)”), Drs. Rajendra and Kim describe experiments aimed at characterizing the effect of exogenous N provided in the form of either ammonium or nitrate on root hair elongation in wild type and csld1 mutant seminal roots. They show that the inhibitory effect of csld1 mutation on root hair elongation can be suppressed by N depletion in the medium. Adding nitrates to the medium leads to normal csld1 root hair growth whereas ammonium leads to root hair growth inhibition. Expression of the AMT1:2 ammonium transporter in the root hairs increased when both nitrate and ammonium were added to the growth medium. Interestingly, AMT1;2expression increased in csld1 mutant root hairs relative to wild type. Furthermore, treatments of CSLD1 over-expressing plants with methylammonium (MeA), a toxic analog of ammonia, suggest that CSLD1 does not participate in nitrogen transport whereas analyses of nitrogen assimilation in another root hair elongation mutant (sndp1) suggest that hair length does not affect nitrogen assimilation. The authors conclude that “(…) CSLD1 is closely involved in nitrogen-dependent root hair elongation and regulation of AMT1;2 expression in rice roots”.
A previous study by the same lab had already reported that this csld1 mutation affects root hair elongation without altering hair initiation, and described an initial molecular characterization of this gene and its family. However, this report provides novel information on a role for exogenous nitrogen to modulate this phenotype. Considering the importance of nitrogen fertilization in plant production, farm economy and its environmental impacts, this study should appeal a broad interest within the scientific community. However, I have several concerns about this document, which I think should be addressed before publication:
1. While the main focus of this manuscript is CSLD1 and its interactions with nitrogen availability and uptake, little information is provided on this gene and other members of its gene family in the Introduction. At the very least, a summary of the information provided in the paper previously published by this lab on this gene and other members of its gene family should be added to the introduction (some of that information is actually included in the discussion, but it seems more appropriate to include it in the introduction such that the readers are aware of the known and unknown before getting into the Results section). On the other hand, a significant amount of background information on root hair mutants in rice and some other plants is included in the results section, and partly duplicated by similar information in the introduction (for instance, see the first paragraph of Results. This information should be removed from the results section (where it distracts from the main data) and consolidated in the Introduction section. The same can be said about the beginning of the last paragraph of the second page of “Results”.
2. Materials and Methods
a. Although the csld1 mutant was obtained from a transposon-mutagenized population (I guess, as this is never stated in this paper), I do not think that a description of the mutagenesis protocol (section 4.1) is necessary here as it was previously reported in the 2007 paper. A thorough description of the mutant is needed, though, better summarizing its properties (null allele? Expression? Phenotypes other than those described in this paper? Was the root hair phenotype rescued by a transgene?) This information should be summarized in the introduction or in Materials and Methods.
b. In section 4.2, the composition of the media used to grow the plants should be specified either with a reference or with the content itself.
c. In section 4.3., which wild type was used to transform the 35S::OsCSLD1 construct? Was the construct also transformed into the csld1 mutant?
d. Under section 4.7., it seems that signal density (number of cells expressing the transgene per length unit of root) should also be measured instead of only focusing on signal intensity (see below).
3. Results
a. In general, I find the Results section difficult to read. It would be helpful to add subtitles separating the different sections, introducing each section with a statement of purpose and following it with a brief conclusion sentence.
b. I would avoid the term “significantly similar”. Statistical analyses test significance of differences, not significance of similarity. “Similar” is sufficient.
c. When analyzing expression of the pCSLD1::GUS construct in CSLD1::Ds rice, the authors mention “ectopic induction of CSLD1 in trichoblasts” (Figure 6). Several questions arise from this experiment:
i. Is CSLD1::Ds/CSLD1::DS the same mutant as csld1? If it is, please use the same terminology throughout, and provide a careful description of the mutant in Materials and Methods and a summary in the Introduction as suggested above.
ii. Figure 6 shows both an increase in signal intensity in root hair cells, and an increase in signal density within the shown section of the root, in the presence of ammonium. Therefore, both signal level per trichoblast and signal density (true significance of “ectopic”) should be QUANTIFIED in this work. Also, a control experiment should be included, testing the same construct under the same conditions in a wild type background.
iii. In the first paragraph of the third page of Results, the interpretation of the MeA treatment experiment seems inappropriate. In view of the results presented in Figure 7C, it seems that csld1 is resistant to MeA whereas the wild type and sndp1 are equally sensitive. Relative root lengths compared to the wild type should be plotted as well to better interpret this result.
d. Figures and Figure legends
i. Figure 1. Please label the panels as indicated in the legend and results sections (A-D)
ii. In Figure 2, the title is an overstatement. Indeed, the data shown in this figure are correlative at best. Double and multiple mutants would be needed to reach this conclusion. Also, in this figure, I would modify the nomenclature of the media, replacing the hyphen between MS and NH4 with a + symbol (to avoid confusion with MS without NH4: MS-NH4)
iii. Figure 3. Again, the title is misleading as mutant root hairs also fail to elongate in the presence of KCl.
iv. Figure 6. Please indicate in the legend the significance of the red bars linking the roots in panels A and C.
v. Figure 7. In the figure legend, please specify that the medium contains MeA in panel D (did it?). Also, a control should be added in this experiment, quantifying ammonium concentrations on MeA-free medium. Finally, as discussed in the previous section, relative root lengths (Control set at 1) should also be plotted to better evaluate sensitivities.
4. The English writing could and should be improved.
Author Response
From,
Prof. Chul Min Kim
Plant Molecular Breeding Laboratory,
Department of Horticulture, Wonkwang University,
54538 Iksan,
Republic of Korea.
Through,
Ms. Nora Huang,
Plants editorial,
MDPI AG, St. Alban-Anlage 66
4052 Basel,
Switzerland.
To,
The editor,
Plants editorial,
MDPI AG, St. Alban-Anlage 66
4052 Basel,
Switzerland.
Dear editor,
I would like to extend my sincere thanks to the reviewers for critically examining the manuscript, which gave us the opportunity to improve the quality of our work by identifying critical points. We were able to make changes and corrections to the manuscript and respond to the comments from the respected editors. Please check the point-by-point response given below.
Review 2
In this manuscript (PLANTS-1989005: OsCSLD1 Mediates NH4+-Dependent Root Hair Growth Suppression and AMT1;2 Expression in Rice (Oryza sativa L.)”), Drs. Rajendra and Kim describe experiments aimed at characterizing the effect of exogenous N provided in the form of either ammonium or nitrate on root hair elongation in wild type and csld1 mutant seminal roots. They show that the inhibitory effect of csld1 mutation on root hair elongation can be suppressed by N depletion in the medium. Adding nitrates to the medium leads to normal csld1 root hair growth whereas ammonium leads to root hair growth inhibition. Expression of the AMT1:2 ammonium transporter in the root hairs increased when both nitrate and ammonium were added to the growth medium. Interestingly, AMT1;2expression increased in csld1 mutant root hairs relative to wild type. Furthermore, treatments of CSLD1 over-expressing plants with methylammonium (MeA), a toxic analog of ammonia, suggest that CSLD1 does not participate in nitrogen transport whereas analyses of nitrogen assimilation in another root hair elongation mutant (sndp1) suggest that hair length does not affect nitrogen assimilation. The authors conclude that “(…) CSLD1 is closely involved in nitrogen-dependent root hair elongation and regulation of AMT1;2 expressions in rice roots”
A previous study by the same lab had already reported that this csld1 mutation affects root hair elongation without altering hair initiation, and described an initial molecular characterization of this gene and its family. However, this report provides novel information on a role for exogenous nitrogen to modulate this phenotype. Considering the importance of nitrogen fertilization in plant production, farm economy and its environmental impacts, this study should appeal a broad interest within the scientific community. However, I have several concerns about this document, which I think should be addressed before publication:
While the main focus of this manuscript is CSLD1 and its interactions with nitrogen availability and uptake, little information is provided on this gene and other members of its gene family in the Introduction. At the very least, a summary of the information provided in the paper previously published by this lab on this gene and other members of its gene family should be added to the introduction (some of that information is included in the discussion, but it seems more appropriate to include it in the introduction such that the readers are aware of the known and unknown before getting into the Results section). On the other hand, a significant amount of background information on root hair mutants in rice and some other plants is included in the results section, and partly duplicated by similar information in the introduction (for instance, see the first paragraph of Results. This information should be removed from the results section (where it distracts from the main data) and consolidated in the Introduction section. The same can be said about the beginning of the last paragraph of the second page of “Results”.
Response: Relevant information was moved to the introduction and more information was added according to reviewer’s comments.
- Materials and Methods
- Although the csld1mutant was obtained from a transposon-mutagenized population (I guess, as this is never stated in this paper), I do not think that a description of the mutagenesis protocol (section 4.1) is necessary here as it was previously reported in the 2007 paper. A thorough description of the mutant is needed, though, better summarizing its properties (null allele? Expression? Phenotypes other than those described in this paper? Was the root hair phenotype rescued by a transgene?) This information should be summarized in the introduction or in Materials and Methods.
Response: Thank you for the comment and we would like to clarify the sources of the materials. The seeds for the experiments were kindly provided my Han’s Lab, Gyeongsang University, Korea and clsd1 gene trap Ds insertion mutant is well characterized by Kim et. al., (2007). We used the same materials and we have cited the source of mutants in mutant materials (4.1).
- In section 4.2, the composition of the media used to grow the plants should be specified either with a reference or with the content itself.
Response: Agreed. Thank you for pointing this out. All the media compositions were followed from Kumar et. al., (2020). The paper has cited in section 4.2 along with other major media references.
- In section 4.3., which wild type was used to transform the 35S::OsCSLD1construct? Was the construct also transformed into the csld1mutant?
Response- Thank you for the question. “Dongjin” was used as the wild type and there were no mutants used for transformation.
- Under section 4.7., it seems that signal density (number of cells expressing the transgene per length unit of root) should also be measured instead of only focusing on signal intensity (see below).
- Results
- In general, I find the Results section difficult to read. It would be helpful to add subtitles separating the different sections, introducing each section with a statement of purpose and following it with a brief conclusion sentence.
Response: Agreed. Thank you for pointing this out. Results section was divided according to reviewer instructions
- I would avoid the term “significantly similar”. Statistical analyses test significance of differences, not significance of similarity. “Similar” is sufficient.
Response: Agreed. Thank you for pointing this out. Wording was corrected according to reviewer comments.
- When analyzing expression of the pCSLD1::GUSconstruct in CSLD1::Dsrice, the authors mention “ectopic induction of CSLD1 in trichoblasts” (Figure 6). Several questions arise from this experiment:
- Is CSLD1::Ds/CSLD1::DSthe same mutant as csld1? If it is, please use the same terminology throughout, and provide a careful description of the mutant in Materials and Methods and a summary in the Introduction as suggested above.
Response: Agreed. Thank you for pointing this out. CSLD1::Ds/CSLD1::Ds is referred as csld1. It was clarified in the text. There were no “CSLD1::Ds/CSLD1::DS” in the text.
- Figure 6 shows both an increase in signal intensity in root hair cells, and an increase in signal density within the showed section of the root, in the presence of ammonium. Therefore, both signal level per trichoblast and signal density (true significance of “ectopic”) should be QUANTIFIED in this work. Also, a control experiment should be included, testing the same construct under the same conditions in a wild type background.
Response: Thank you for your kind suggestion. We also like to explore the possibility to quantify the signal level. However, It’s difficult to obtain a transposon insertion mutant without GUS construct. Please refer Kim et. al., (2007) for the information on the construct and mutant information. Quantification of signal is highly variable with time and conditions. Even if the quantification is performed, the results can be unreliable.
iii. In the first paragraph of the third page of Results, the interpretation of the MeA treatment experiment seems inappropriate. In view of the results presented in Figure 7C, it seems that csld1 is resistant to MeA whereas the wild type and sndp1 are equally sensitive. Relative root lengths compared to the wild type should be plotted as well to better interpret this result.
Response: Agreed. Thank you for pointing this out. Text was rewritten according to the reviewer’s comments.
- Figures and Figure legends
- Figure 1. Please label the panels as indicated in the legend and results sections (A-D)
Response: Agreed. Thank you for pointing this out. Figures Labelled according to the reviewer comments.
- In Figure 2, the title is an overstatement. Indeed, the data shown in this figure are correlative at best. Double and multiple mutants would be needed to reach this conclusion. Also, in this figure, I would modify the nomenclature of the media, replacing the hyphen between MS and NH4 with a + symbol (to avoid confusion with MS without NH4: MS-NH4)
Response: Agreed. Thank you for pointing this out. Figures were corrected and text was simplified according to reviewer comments.
iii. Figure 3. Again, the title is misleading as mutant root hairs also fail to elongate in the presence of KCl.
Response: Agreed. Thank you for pointing this out. Text was corrected according to reviewer comments.
- Figure 6. Please indicate in the legend the significance of the red bars linking the roots in panels A and C.
Response: Agreed. Thank you for pointing this out. Red bars indicate the regions which were used to get magnified images. It was cited in figure legends according to reviewer comments.
- Figure 7. In the figure legend, please specify that the medium contains MeA in panel D (did it?). Also, a control should be added in this experiment, quantifying ammonium concentrations on MeA-free medium. Finally, as discussed in the previous section, relative root lengths (Control set at 1) should also be plotted to better evaluate sensitivities.
Response: Thank you for your question. No MeA was added when quantifying ammonium content. It was a typing mistake.
- The English writing could and should be improved.
Response: Thank you for your concern. English correction was already performed, please kindly find the attached English correction proof at the end of the manuscript.
We have made relevant corrections with track changes and attached the corrected manuscript. However, we would like to inform that the comment section 3.a (GUS construct), the suggestion from the reviewer 2 is not achievable in near future. Other comments were very critical, and we made good improvements accordingly. We are eagerly waiting for the response from the respectable reviewers soon.
Sincerely,
Chul Min Kim

Round 2
Author Response
From,
Prof. Chulmin Kim
Plant Molecular Breeding Laboratory,
Department of Horticulture, Wonkwang University,
54538 Iksan,
Republic of Korea.
Through,
Ms. Nora Huang,
Plants editorial,
MDPI AG, St. Alban-Anlage 66
4052 Basel,
Switzerland.
To,
The editor,
Plants editorial,
MDPI AG, St. Alban-Anlage 66
4052 Basel,
Switzerland.
25.11.2022
Dear editor,
I would like to extend my sincere thanks to the reviewer 2for further examining the manuscript, which gave us more elaborate corrections and updates to improve the quality of our work. We made critical changes and corrections to the manuscript and respond to the comments from the respected reviewer 2. Please check the point-by-point response given below.
Reviewer 2
This revised version of the manuscript by Drs Rajendran and Kim (PLANTS-1989005) addresses some of the concerns and suggestions made on the previous draft, including: 1) Transferring some information on the mutants from the Results section into the Introduction, 2) Adding references and clarifications on the mutants and wild type genotypes used in these analyses, and 3) Correcting several over-statements and confusing sentences both in the texts and in the figure legends. This somewhat improved the quality of this draft. However, I still think this manuscript is not sufficiently “cleaned up” to warrant its publication at this time, for the following reasons:
- Comment: The introduction remains confusing and difficult to read. For instance, there is substantial redundancy between the third and last paragraphs of the introduction. These paragraphs describe the contribution of cellulose synthase-like enzymes in wall metabolism and cell growth. However, they are separated from each other by another paragraph that describes several additional mutations affecting root hair development. This disruption makes it difficult to truly understand what is already known about the role of CSLDs in root hair development. The introduction should be restructured to consolidate these two paragraphs, remove redundancy and better streamline the information necessary to better understand the research reported in this manuscript.
- Response: Agreed, Thank you for the suggestion. Introduction has been revised and sectioned accordingly.
- Comment: There is a lack of concordance between the information provided in the first section of Results and the legends to Figures 1 and 2. Are the media described in these figures lacking nitrate and/or ammonium (as suggested in the Introduction), or do they contain these compounds (as suggested in the figure legends)? Considering that the information presented in these two figures constitutes a premise for the entire paper, it is quite surprising (and frustrating, to be honest!) that the authors did not pay more careful attention to carefully describing the media used in these studies.
- Response: Agreed, we have clarified and corrected the media composition in figures according to reviewer comments.
- Comment: Section 2.2 starts with an introductory sentence on AMT1;2 that I actually do not understand. Section 2.4. also starts with a description of the same group of three ammonium transporters (AMT1;1, AMRT1;2 and AMT1;3), but with many more details than section 2.2. In fact, I believe this indicates the complete information on this family of ammonium transporters should be transferred to the Introduction section, and the two Results section should be introduced with a short sentence summarizing what is currently known and relevant to each section.
- Response: Agreed, We have revised above mentioned sections and introduction was updated accordingly.
- Comment: Still in section 2.2., I still do not understand what the authors mean by “ectopic induction of the CSLD1 gene in trichoblasts”. There is really a need to better illustrate this result. Considering that trichoblasts and atrichoblasts are “arranged randomly in rice to facilitate plastic response under alternating surrounding conrditions” (excerpt from the Introduction), the result would imply a higher density of trichoblasts in the presence of ammonium or nitrate. Is this the case? This figure should be modified to show better resolution images clearly displaying epidermal cell borders, blue staining, and quantification data addressing the frequency of blue-stained cells in a defined section of the root, and cell sizes.
- Response 1: Agreed, some readers may misunderstand the meaning of “Ectopic” expression of CSLD1. The text mentioned the expression of GFP due to the Ds Text was corrected and clarified.
- Response 2: Disagreed. CSLD1 is not characterized for cell fate determination but cellular morphology (Cellulose synthase-like gene), because overexpression of CSLD1 did not increase the number of trichoblast cells (Kim al., 2007). Figure 6 illustrates the representative expression pattern of Ds inserted CSLD1 indicated by GUS. Moreover, Trichoblasts and atrichoblasts are hard to identify in young stages before the initiation or root hair. As explained in previous review, “3.c.ii. Quantification of signal is highly variable with time and conditions. Even if the quantification is performed, the results can be unreliable”. Quantification of cells and cell sizes will not have major impact in the results because CSLD1 is not related to cell cycle or cell fate determination process.
- Comment: In lines 441-442, the authors mention: “(…)AMT1;3 showed a slight increase in expression in the csld1 mutant background (Figure 5a)”. This seems like an over-statement because the statistical analysis reported in that figure indicates that wild type and mutant seedlings belong to the same statistical group (c) for AMT1;3 expression.
- Response: Agreed, Results and discussion was changed according to reviewer comments.
- Comment: The legends of figures containing quantitative data should include information on the numbers of plants included in the analyses.
- Response: Agreed, Replicate numbers added.
- Comment: The data shown in Supplemental Figure 1 suggest strong resistance of csld1 to MeA, especially in view of the fact that mutant seminal roots seem shorter than wild type in the absence of MeA. A third graph should be added to this figure, illustrating relative root length for all genotypes (normalized to lengths on 0 mM MeA). Furthermore, statistical significance symbols should be added to this graph.
- Response: Agreed. We have changed the line graph to bar graph which shows more detailed visualization of data. Statistical significance added and relative length graph added. However, the relative length graph may be redundant because it shows similar patterns with the actual data.
- Comment: Several sentences should be re-written because they are either misleading or simply very difficult to understand. For instance, I do not understand what the authors meant when writing the following sentences:
- Response: Thank you for the comment. However, we have done English correction from a reputed firm. However, the texts were amended with simplified English for understanding. Moreover, below mentioned line numbers are from the materials and methods sections from our recently revised draft. We revised all possible texts we can track with provided information. Please kindly refer the recently revised draft and kindly specify if you are referring to the initial draft.
- Lines 437-438.
- Response: We couldn’t find the exact text. We have revised all parts. Thank you for your consideration.
- Lines 456-459 (see above).
- Response: We couldn’t find the exact text. We have revised all parts. Thank you for your consideration.
- The last paragraph of the Results section (lines 486-568: I don’t know why the numbers shift from the 400s to the 500s at page break here) is very difficult to understand because it starts with an introductory sentence describing an experiment that makes use of CSLD1: OX lines, then shifts to results one experiments that use AMT1;2:OX over-expressing lines, before getting back to a description of results for CSLD1:OX lines. This is highly confusing! This paragraph should be re-written to better separate the experiments involving these two distinct sets of over-expressing lines.
- Response: Thank you for your comments. Track changes enabled documents will contain line numbers until the track changes are accepted or rejected. Results were separated and explained separately according to the reviewer comments.
- Lines 598-599
- Response: We couldn’t find the exact text. We have revised all parts. Thank you for your consideration.
We have made relevant corrections with track changes and attached the corrected manuscript. However, we would like to inform that the comment section 4. (Related to Figure 6.), the suggestion from the reviewer 2 has been previously clarified as unreliable. Other comments were very critical and we made good improvements accordingly. We would like to extend our thanks and gratitude to the reviewer 2. We are eagerly waiting for the response from the respectable reviewers soon.
Sincerely,
Chul Min Kim

Round 3
Reviewer 2 Report
In this new revision of their manuscript (PLANTS-1989005), Drs Rajendran and Kim adequately address the comments and suggestions I made of their previous draft, thereby improving the quality of the manuscript. Importantly, they deleted the concept of “ectopic expression” when referring to the expression of CSLD-GUS in csld1 mutants relative to wild type, replacing it by the more appropriate concept of “increased expression”. Overall, the data are interesting and important, warranting publication in Plants.
I have only very minor suggestions on this draft:
1. Line 74: Please replace “forms” with “form”;
2. Line 108: Please replace “plays” with “play”;
3. Line 110: Please insert “are” between “AMT1;3” and “known”;
4. Line 121: Please replace “was” with “were”;
5. Line 122: Please replace “regulates” with “regulate”;
6. Figure 1. Please insert the scale bars in the figures, as referenced in the legend;
7. Lines 186-187: Please replace “than the standard” with “relative to the standard”;
8. Legend to Figure5: How many seedlings were analyzed in this experiment (N=?)?
9. Line 263: Please consider deleting “However” because it is too close from the previous one in the text (line 259), bringing unnecessary confusion;
10. Line 269: Please replace “was” with “were”;
11. Line 270: Please consider replacing “Strongly observed in trichoblasts” with “increased in trichoblasts relative to control conditions”;
12. Figure 6. Please add a scale bar to panel E as it illustrates a different root relative to the other panels for this figure. Also, please indicate the size of the scale bars in the figure Legend;
13. Line 284: Please add a reference supporting the claim made in the first sentence;
14. Line 327: Please consider replacing “and” at the end of by line with “whereas”;
15. Line 330: Please consider replacing “than that of the wild type” with “relative to wild type grown in the presence of MeA”;
16. Line 331-334. Please consider replacing this sentence with something like: “This observation was further validated when csld1 and CSLD1:OX seedlings were grown in the presence of MeA. In this case, csld1 seminal roots remained longer than wild type and CSLD1:OX roots when exposed to increasing concentrations of MeA (Figure S1”.
17. Line 354: Please delete “However”, because two successive sentences start with the same word.
18. Lines 377-378: This sentence does not make much sense as is. Please consider replacing it with something like: “However, the presence of NH4+ led to strong suppression of root hair length in csld1 mutants.”
19. Line 399: Please replace “between” with “for”.

Author Response
From,
Prof. Chulmin Kim
Plant Molecular Breeding Laboratory,
Department of Horticulture, Wonkwang University,
54538 Iksan,
Republic of Korea.
Through,
Ms. Nora Huang,
Plants editorial,
MDPI AG, St. Alban-Anlage 66
4052 Basel,
Switzerland.
To,
The editor,
Plants editorial,
MDPI AG, St. Alban-Anlage 66
4052 Basel,
Switzerland.
07.12.2022
Dear editor,
Response for the round 3 revisions of manuscript: plants-1989005
I would like to extend my sincere thanks to the reviewer 2 for deeply examining the manuscript, which gave opportunity to clean-up the manuscript. We made critical corrections to the manuscript and respond to the comments from the respected reviewer 2. Please check the point-by-point response given below.
Reviewer 2
In this new revision of their manuscript (PLANTS-1989005), Drs Rajendran and Kim adequately address the comments and suggestions I made of their previous draft, thereby improving the quality of the manuscript. Importantly, they deleted the concept of “ectopic expression” when referring to the expression of CSLD-GUS in csld1 mutants relative to wild type, replacing it by the more appropriate concept of “increased expression”. Overall, the data are interesting and important, warranting publication in Plants.
I have only very minor suggestions on this draft:
- Comment: Line 74: Please replace “forms” with “form”;
- Response: Agreed, Thank you for the correction. word changed.
- Comment: Line 108: Please replace “plays” with “play”;
- Response: Agreed, Thank you for the correction. word changed.
- Comment: Line 110: Please insert “are” between “AMT1;3” and “known”;
- Response: Agreed, Thank you for the correction. word insterted.
- Comment: Line 121: Please replace “was” with “were”;
- Response: Agreed, Thank you for the correction. word changed.
- Comment: Line 122: Please replace “regulates” with “regulate”;
- Response: Agreed, Thank you for the correction. word changed.
- Comment: Figure 1. Please insert the scale bars in the figures, as referenced in the legend;
- Response: Agreed, Thank you for the correction. Scale bars added.
- Comment: Lines 186-187: Please replace “than the standard” with “relative to the standard”;
- Response: Agreed, Thank you for the correction. word changed.
- Comment: Legend to Figure5: How many seedlings were analyzed in this experiment (N=?)?
- Response: 3 biological replicates and 2 technical replicates for each. This is indicated in materials and methods (section 4.8).
- Comment: Line 263: Please consider deleting “However” because it is too close from the previous one in the text (line 259), bringing unnecessary confusion;
- Response: Agreed, Thank you for the correction. word changed.
- Comment: Line 269: Please replace “was” with “were”;
- Response: Agreed, Thank you for the correction. word changed.
- Comment: Line 270: Please consider replacing “Strongly observed in trichoblasts” with “increased in trichoblasts relative to control conditions”;
- Response: Agreed, Thank you for the correction. sentence changed.
- Comment: Figure 6. Please add a scale bar to panel E as it illustrates a different root relative to the other panels for this figure. Also, please indicate the size of the scale bars in the figure Legend;
- Response: Response: Agreed, Thank you for the correction. Scale bars added.
- Comment: Line 284: Please add a reference supporting the claim made in the first sentence;
- Response: Agreed, Thank you for the correction. Reference added.
- Comment: Line 327: Please consider replacing “and” at the end of by line with “whereas”;
- Response: Agreed, Thank you for the correction. word changed.
- Comment: Line 330: Please consider replacing “than that of the wild type” with “relative to wild type grown in the presence of MeA”;
- Response: Agreed, Thank you for the correction. sentence changed.
- Comment: Line 331-334. Please consider replacing this sentence with something like: “This observation was further validated when csld1and CSLD1:OX seedlings were grown in the presence of MeA. In this case, csld1 seminal roots remained longer than wild type and CSLD1:OX roots when exposed to increasing concentrations of MeA (Figure S1”.
- Response: Agreed, Thank you for the correction. sentence changed.
- Comment: Line 354: Please delete “However”, because two successive sentences start with the same word.
- Response: Agreed, Thank you for the correction. word removed.
- Comment: Lines 377-378: This sentence does not make much sense as is. Please consider replacing it with something like: “However, the presence of NH4+ led to strong suppression of root hair length in csld1”
- Response: Agreed, Thank you for the correction. However, we could not locate the exact lines because of template changes. We hope the above mentioned suggestion is strongly inferred in the figures and results.
- Comment: Line 399: Please replace “between” with “for”.
- Response: Agreed, Thank you for the correction. word changed.
We have made relevant corrections with track changes and attached the corrected manuscript. comments were very critical and we made good improvements accordingly. We would like to extend our thanks and gratitude to the reviewer 2. We are eagerly waiting for the response from the respectable reviewers soon.
Sincerely,
Chul Min Kim
